# Application and Development Progress of Cr-Based Surface Coatings in Nuclear Fuel Element: I. Selection, Preparation, and Characteristics of Coating Materials

**Huan Chen, Xiaoming Wang and Ruiqian Zhang \***

Nuclear Power Institute of China, Chengdu 610200, China; npicchh@163.com (H.C.); npicwxm@163.com (X.W.)
**\*** Correspondence: zhang_ruiqian@126.com or fanyingduiranliaojicailiaozhongdianshiA@npic.ac.cn

**Abstract:** To cope with the shortcomings of nuclear fuel design exposed during the Fukushima Nuclear Accident, researchers around the world have been directing their studies towards accident-tolerant fuel (ATF), which can improve the safety of fuel elements. Among the several ATF cladding concepts, surface coatings comprise the most promising strategy to be specifically applied in engineering applications in a short period. This review presents a comprehensive introduction to the latest progress in the development of Cr-based surface coatings based on zirconium alloys. Part I of the review is a retrospective look at the application status of zirconium alloy cladding, as well as the development of ATF cladding. Following this, the review focuses on the selection process of ATF coating materials, along with the advantages and disadvantages of the current mainstream preparation methods of Cr-based coatings worldwide. Finally, the characteristics of the coatings obtained through each method are summarized according to some conventional performance evaluations or investigations of the claddings. Overall, this review can help assist readers in getting a thorough understanding of the selection principle of ATF coating materials and their preparation processes.

**Keywords:** zirconium alloy; accident-tolerant fuel; ATF cladding material; surface coating; coating preparation

## 1. Introduction

The nuclear fuel element is the core component of a nuclear reactor that plays a role in releasing heat and tolerating fission gas [1]. The existing nuclear fuel elements of light water reactors (LWRs, including pressurized water reactors (PWRs) and boiling water reactors (BWRs)) are composed of nuclear fuel pellets of $UO_2$ and cladding materials of zirconium alloy. During the Fukushima Daiichi Accident, zirconium cladding underwent high-temperature chemical reactions with steam under the circumstance that the cooling capacity of the system was lost due to an earthquake and the resulting tsunami. The reactions that followed then triggered multiple hydrogen explosions, which finally caused severe radiological hazards [2]. This disaster revealed the weaknesses of the existing $UO_2$-zirconium alloy fuel configuration in resisting severe accidents, such as loss-of-coolant accidents (LOCAs). In response to the design flaws exposed in the Fukushima Nuclear Accident, the research focus of nuclear fuel has globally shifted towards a fuel with a specific ability to tolerate accidents, i.e., accident-tolerant fuel (ATF) [3–11]. This unique fuel design is expected to improve the safety of reactors and fuel.

Compared with current $UO_2$-Zr fuel, the new accident-tolerant fuel enables a reactor core to remain intact for a rather long time when it loses its external cooling supply and is not given any intervention from the operator [3]. One of the core tasks of the ATF design is to develop cladding



materials with outstanding performance. The cladding should have the relatively good ability to resist irradiation and corrode uniformly, a relatively small cross-section for the absorption of a neutron, and long-term stable thermal conductivity, as well as the ability to resist high-temperature oxidation and the ability to retain good mechanical properties at the temperature of accident conditions, thereby sustaining the structure of the nuclear fuel element that remains intact [10–12]. Surface coatings are the technological trend of modifying zirconium alloy to enhance accident tolerance without causing significant changes to the existing $UO_2$/Zr alloy cladding fuel design that has been extensively used in commercial water-cooled reactors, thereby holding tremendous application prospects [13].

Since the inception of the ATF concept and its development, many countries have accumulated extensive experience and achievements in the research and development of zirconium alloys with Cr-based coatings [14–25]. However, existing literature and reports have not been able to establish a systematic framework. More specifically, researchers from the non-nuclear field do not have a clear understanding of the application background of ATF coating materials, which impedes the development of related research. To fill this gap, a more detailed critical review is presented that focuses on the latest progress in the development of Cr-based surface coatings on zirconium alloys employed in accident-resistant fuel elements. This review is divided into two sections: Part I is devoted to the selection, preparation, and characteristics of coatings, and Part II discusses the current status and shortcomings of performance studies on zirconium alloys with Cr-based coatings.

In Part I of this review, the current status of the application of zirconium cladding, along with the proposal and development of ATF cladding, are addressed to clarify the original intention of applying coating technology to nuclear fuel. Then, the section focuses on the selection process and preparation methods of ATF coating materials. Specifically, it clarifies the adaptability of various coating systems under the operating environment of a reactor, the advantages and disadvantages of the current mainstream preparation methods of Cr-based coatings around the world, and the characteristics of coatings prepared by each method.

## 2. Current Status of Application of Zirconium Alloy Claddings

Nuclear power plants are facilities that utilize the energy released from nuclear fission reactions that take place in the reactor and then generate electricity. Nuclear fission occurs in the reactor core. In LWRs, fuel pellets containing oxides of fissile elements (usually $UO_2$) are loaded into a cladding tube. The fuel pellets are then compacted by a hold-down spring. Finally, a single fuel rod is formed by welding the upper and lower end plugs to the corresponding ends of the cladding tube loaded with fuel pellets. A group of fuel rods and guide thimble tubes with nozzles and grid spacers are bundled together to form a fuel assembly (fuel element). Specific fuel assemblies are then packed together to build the core of a nuclear reactor. Figure 1 schematically presents the composition of a single fuel assembly in a pressurized water reactor (PWR) [8].

Over the past 65 years, nuclear power development has indicated that nuclear safety is a prerequisite for the successful utilization of nuclear technology. Cladding is one of the essential components that enables the fuel elements to remain intact and ensures the safety of a nuclear power plant. As a second safety shield in a nuclear power plant [1], cladding prevents the dissipation of fission products, avoids the corrosion of fuel by the coolant, and effectively dissipates thermal energy. Among all the structural materials in a reactor core, cladding operates under the severest operating conditions. It experiences both high temperature and high pressure during core operation. Besides, the inner surface of the cladding withstands the attack of nuclear fission products, intense neutron irradiation and pellets swelling, resulting in the pellet-cladding interaction (PCMI), while the outer surface is subject to the corrosion and erosion of cooling water, as well as the abrasion caused by fretting wear between the cladding and the assembly grid or even the debris in the cooling water [1]. Therefore, the design of a reactor core has stringent requirements in order to demonstrate the performance of its cladding material. Hence, the design of a reactor needs to be comprehensively considered and balanced from various aspects, such as neutron economy, irradiation stability, thermal conductivity,

mechanical strength, and chemical compatibility [1,26–28]. Additionally, to improve the utilization efficiency of thermal energy, boost the economic competitiveness of nuclear power, and minimize the amount of nuclear waste, it is essential to achieve a higher burnup by raising the operating power, extending the refueling cycle, and extending the lifespan of the reactor [26,29–31]. An increase in the operating temperature and irradiation exposure due to these measures would undoubtedly raise the requirements of the performance of the cladding.

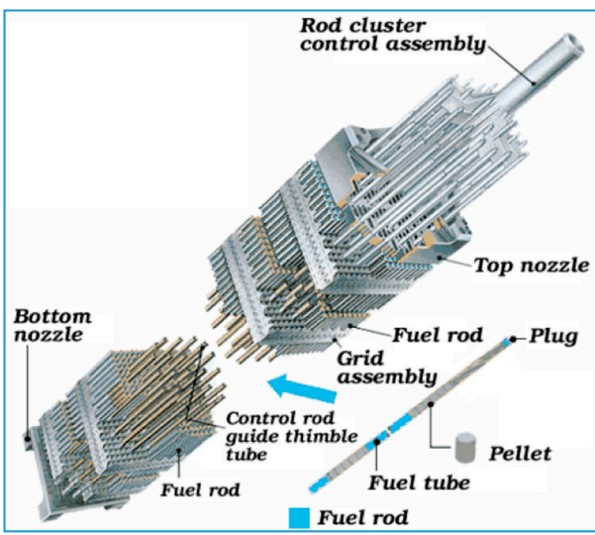

**Figure 1.** The schematic diagram of a typical pressurized water reactor (PWR) fuel assembly. Reproduced from [8] with copyright permission from Elsevier.

Zirconium alloy cladding is being widely used in water-cooled reactors due to its unique performance advantages and the development of manufacturing technology during the last few decades [1,8,26–28,30–32]. In the history of zirconium alloy development, researchers from the US were the first to add Sn, Fe, Cr, and Ni as alloying elements into zirconium to improve its corrosion resistance and mechanical properties. Two zirconium alloys, i.e., Zr-2 and Zr-4, were developed. Following this, researchers from the former Soviet Union developed the E110 alloy by adding Nb to the zirconium alloy. With the development of nuclear reactor technology, there is an urgent need for the continuous improvement of zirconium alloys to satisfy the ever-growing demand of burnup, which drives the optimization of existing zirconium alloys and the development of new alloys. To accomplish these goals, several advanced zirconium alloys, such as ZIRLO, OPT ZIRLO, and AXIOM in the US, M5 in France, E635 in Russia, HANA in South Korea, MDA in Japan, and N36 in China, have been developed via adjusting types of alloying elements (Sn, Nb, Fe, Cr, Ni, Cu, V, Si, etc.) and improving their heat treatment processes [8,30–32]. (E110, ZIRLO, AXIOM, M5, E635, HANA, MDA and N36 are trademarks of nuclear grade zirconium alloy in various countries)

The performance of zirconium alloy cladding during the fuel cycle is summarized in Table 1 [33,34]. Under normal operating conditions, the use of commercial zirconium alloy claddings can usually meet the significant performance requirements of existing reactors for resistance to corrosion, creep, irradiation growth, and abrasion [35–37]. However, their inherent water-side corrosion and accompanying embrittlement caused by hydrogen absorption are still the critical factors that cause cladding failure and limit a further increase in nuclear fuel lifespan and burnup [1,13,26,30,31,38]. Notably, under accident conditions, the rapid oxidation of zirconium alloy cladding in high-temperature steam (releasing a lot of heat and flammable hydrogen), balloon/burst (secondary hydrogenation), and the accompanying reduction in the ductility seriously threaten the integrity of the cladding and even the core [39–41]. Once the situation is out of control, the failure of the cladding tube causes severe accidental consequences (such as the Fukushima Nuclear Accident [2]) that pose considerable risk

to the safety of the reactor. Additionally, hydride reorientation is an issue that may cause the failure of zirconium alloy cladding during the spent fuel storage [42]. Though the performance of cladding under normal operating conditions has been further improved through the development of advanced zirconium alloys, its performance improvement at high temperatures is still unsatisfying due to the limitations of the zirconium alloy material itself [8,11,13,31]. It is difficult to make a breakthrough merely by the addition of trace alloying elements or by improving the heat treatment process, as these technical routes have bottlenecks [11,31]. Therefore, to further improve the operating performance, economic efficiency, and safety of water-cooled reactors, it is necessary to explore and develop new cladding materials that have superior service performance and accident resistance than the current commercial zirconium alloy cladding.

**Table 1.** Performance of the commercial zirconium alloy cladding during the fuel cycle. Adapted from [34] with copyright permission from Elsevier. More information found in [11,35–38,42–44].

| Performance Parameters | | Commercial Zirconium Alloy Cladding | Limit Values |
|---|---|---|---|
| Manufacturability | | Optimized | - |
| Normal operating conditions | Corrosion | No problem | Corrosion depth < 10% of cladding thickness [35,37] |
| | Creep | No problem | Hoop strain < 1% [35–37] |
| | Irradiation growth | No problem | Shoulder gap closure |
| | Wear resistance | No problem | Wear damage < 10% of cladding thickness [37] |
| | PCMI/PCI [36,38] | Failure related to hydrides (corrosion and hydrogen absorption) | - |
| DBA | LOCA [36,38] | Severe oxidation and hydrogen generation; Balloon and burst | PCT < 1204 °C; ECR < 17% [37,43] |
| | RIA [36,38] | PCMI failure related to hydrides | - |
| BDBA [11,36,38] | | Severe oxidation and hydrogen generation; Melting point problem | - |
| Spent fuel storage [42,44] | | Hydrogen-induced embrittlement behavior Hydride reorientation problem | Fuel-cladding interface temperature: drying < 570 °C, storage < 400 °C; Clad hoop stress < 90 MPa [42] |

Note: PCMI: pellet–cladding mechanical interaction; PCI: pellet–cladding interaction; DBA: design-based accident; BDBA: beyond design-based accident; LOCA: loss of coolant accident; RIA: reactivity-initiated accident; PCT: peak cladding temperature; ECR: equivalent cladding reacted.

## 3. Proposal and Development of ATF Cladding

The Fukushima Nuclear Accident in Japan triggered the shifting of attention to the safety of the nuclear power plants around the world and promoted the development of new nuclear technologies. The Westinghouse Electric Corporation in the US proposed the concept of ATF for the first time in 2003 [8]. ATF was designed to improve the passive safety property of water-cooled reactors, increasing their safety allowance and reducing their dependence on active cooling [8]. After the Fukushima Nuclear Accident, the US was the first to raise an R&D plan of ATF to the government level for an acceleration in the development and deployment of ATF. It has been widely acknowledged that compared with the currently used standard $UO_2$-zirconium alloy system in the nuclear industry, ATF is an advanced fuel system that can withstand the loss of the active cooling of the reactor core under higher

temperatures for a longer time. It can provide longer "processing time" under accident conditions while maintaining or improving fuel performance during its regular operation [3]. Considering that countries are further strengthening nuclear safety, ATF has internationally become a hot research area in the field of fuel in the post-Fukushima era.

In the design of ATF, the accident-tolerant properties of claddings refer to enhanced resistance to high-temperature oxidation (more than 100 times higher that of zirconium alloy cladding) and a high-temperature strength equal to or better than zirconium alloys [11]. As seen in Figure 2 [11], ATF cladding can extend the time for the occurrence of the ballooning–bursting of a fuel rod, and it can decrease the rising rate of cladding temperature, thereby dramatically delaying the progress of an accident. Cheng et al. [45] analyzed and evaluated on the achievements of ideal ATF cladding in improving the resistance of a fuel to damage in a severe accident (with the hypothesis that ATF cladding can retain its full resistance to a reaction with steam before its melting behavior at 2500 °C). Analysis results have shown that for typical PWRs and BWRs, employing an ideal ATF cladding can lead to 28 and 5 h gains, respectively. Meanwhile, under real operating conditions, if fuel rods are resistant to steam corrosion at 1200 and 1500 °C, then PWRs can obtain 5 and 20 h gains, respectively.

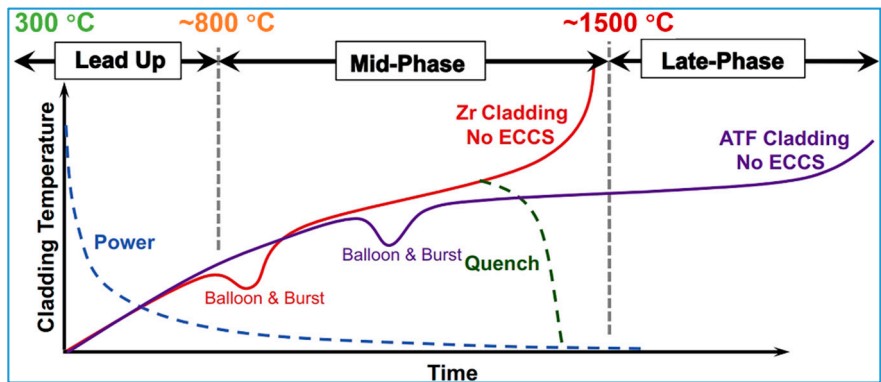

**Figure 2.** Potential delaying effect of accident-tolerant fuel (ATF) cladding on the development of a severe accident. Reproduced from [11] with copyright permission from Elsevier.

For an ATF cladding, the most fundamental requirement is to reduce the oxidation rate of the cladding in a high-temperature steam environment [10–12,46]. As a result, the generation of heat and hydrogen is dramatically reduced. Hence, the burden on the emergency core cooling system (ECCS) during severe accidents could be eased [46]. In the design and development of high-temperature alloys (i.e., superalloys), three conventional protective oxide films, i.e., $Cr_2O_3$, $Al_2O_3$, and $SiO_2$ [47–50], can be used to protect the underlying materials. These oxides can provide potential protection since the diffusion rates of oxygen through the oxide layers are low enough to ensure a low oxidation rate, which forces the oxides to grow slowly and eventually form continuous protective oxide layers. As shown in Figure 3, the conventional materials that can form the above-mentioned three protective oxide films are SiC, FeCrAl, and 310SS (one kind of code of stainless steels), and their parabolic oxidation rate constants are reduced by approximately two orders of magnitude compared to Zr-4 [46,51].

An ATF cladding is mainly made from materials that can form one of the three protective oxide films, as mentioned above. Two major technical routes have been established around the world. One route is to develop alternative materials, such as $SiC_f$/SiC composites [6,10,52–56], FeCrAl stainless steels [57–61], Mo alloys [45,62–65], and MAX phases (the family of layered ternary compunds ($M_{n+1}AX_n$) in which M is an early transition metal, A is an A-group element and X is either carbon or nitrogen.) [51,56,66–68], which can completely replace zirconium alloys. The other route is to prepare coatings on the surface of the existing zirconium alloy claddings [13,33,34,69–74]. According to the ATF R&D plans of various countries in the world [75–78], different research institutes have focused on

the R&D directions of ATF claddings. The leading central institutes of each country and their focuses on ATF claddings are summarized in Table 2.

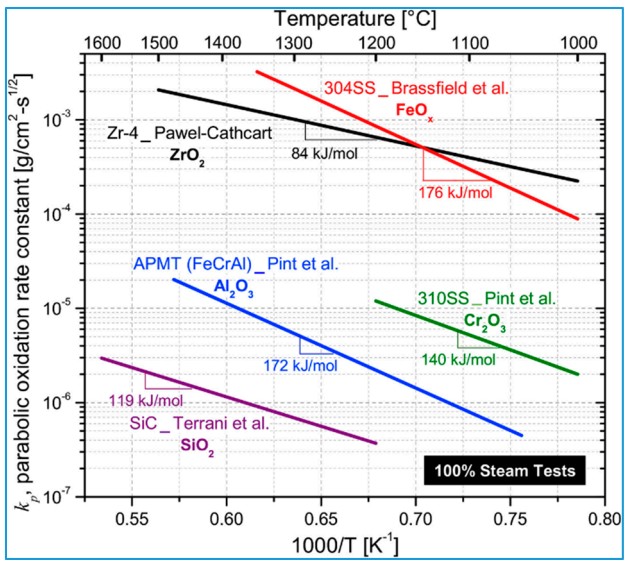

**Figure 3.** The parabolic oxidation rate of various cladding materials and their resulting oxides in high temperature steam. Reproduced from [46] with copyright permission from Elsevier.

**Table 2.** Leading institutes of each country and their preferred research directions [75–78].

| Leading Institute | Abbreviation | Country | Preferred Research Directions |
|---|---|---|---|
| Oak Ridge National Laboratory | ORNL | US | FeCrAl, SiC$_f$/SiC, MAX phases, coated Zr alloy |
| Los Alamos National Laboratory | LANL | US | FeCrAl, Mo alloy |
| Westinghouse Electric Corporation | WHE | US | Coated Zr alloy, SiC$_f$/SiC |
| General Electric Company | GE | US | FeCrAl |
| Electric Power Research Institute | EPRI | US | Mo-alloy |
| Areva S.A. | AREVA | France | Coated Zr alloy |
| French Alternative Energies and Atomic Energy Commission | CEA | France | Coated Zr alloy, SiC$_f$/SiC |
| Korea Atomic Energy Research Institute | KAERI | South Korea | Coated Zr alloy, SiC$_f$/SiC |
| Japan Atomic Energy Agency | JAEA | Japan | FeCrAl, SiC$_f$/SiC |
| Nuclear Power Institute of China | NPIC | China | Coated Zr alloy, FeCrAl, SiC$_f$/SiC |

Among the materials, new alternative cladding materials have several potential advantages that are helpful for improving the accident resistance of fuel cladding. For example, SiC$_f$/SiC composites have excellent high-temperature mechanical properties, chemical properties, and neutron economy [10,31,46,56]. The FeCrAl alloy has an excellent corrosion/oxidation resistance and a relatively superior high-temperature strength under normal and accident conditions [10,31,46,56]. Additionally, it is not difficult to manufacture this alloy. Mo, or Mo alloys, features an ultra-high melting point (2623 °C), excellent high-temperature strength, and high-temperature creep behavior [10,31,45]. MAX phases have a strong stiffness, good thermal conductivity, and high resistance to chemical attacks, and high-temperature oxidation [51,56,66–68]. However, these new materials also have

some specialproblems, such as the corrosion/dissolution of SiC that occurs under normal operating conditions of water-cooled reactors [10,46], the high tritium permeability of FeCrAl [10,46], and the high volatility of Mo under high-temperature oxidation conditions [10,45]. It is difficult to eliminate the shortcomings of the materials themselves when only a single material is used, and coating technology should still be employed when necessary [10,45]. More importantly, there is evidence that all of these materials show the obvious degradation of their mechanical properties under neutron irradiation. For example, the irradiation damage of neutrons and light gas (He and H) creates bubbles, cavities, blisters, microcracks, etc., thus causing mechanical problems such as creep, fracture, and embrittlement [45,79–86]. All the above issues mean that the use of these materials to replace Zr alloys is still faced with great application risks and technical challenges, and more efforts are needed to solve them. Additionally, due to the large difference in the neutron physical properties between these materials and Zr alloys, it is necessary to make major engineering redesign decisions for cores. For example, the higher neutron absorption of FeCrAl and Mo can lead to problems of an increased fuel enrichment and a decreased cladding wall thickness [31,45].

At this stage, it is difficult to judge which material or technical route is safer, but we can be sure that the cladding concept of completely replacing zirconium alloy has a greater uncertainty and should be a long-term goal. In contrast, using zirconium alloys with surface coatings provides improved performance without requiring significant changes to the current $UO_2$/zirconium alloy cladding fuel design [13,31,46]. This is a less uncertain and more conservative technical solution, which makes it a more possible short-term goal. Therefore, coatings on zirconium alloy surfaces is the preferred technical route to realize engineering application in nuclear industry more quickly.

## 4. Development of ATF Coating Materials

The most straightforward way to transition from conventional to ATF claddings is to deposit a protective coating on the surface of a Zr alloy cladding in a current water-cooled reactor fuel system. The main advantages of the coating technology can be understood from two practical aspects: The application of the coating is cost-effective because existing commercial Zr alloys and manufacturing facilities can be used, and they can be gradually deposited on the current cladding material without substantially changing the physical state of the core [13,31,46].

Considering the definition of ATF, to maintain (if not to improve) fuel performance under normal operating conditions and enhance the accident resistance and risk compatibility of the cladding, the coated ATF claddings should meet the following requirements [13]: (i) a good adhesion to the substrate and abrasion resistance along with sufficient ductility; (ii) good resistance to hydrothermal corrosion in PWR or BWR chemistry; (iii) good irradiation stability in neutron irradiation environment of the water-cooled reactor; (iv) enhanced resistance to high-temperature steam oxidation under accident conditions; (v) no adverse effects of the coating on the Zr alloy matrix during the preparation process and throughout the fuel life. The above requirements allow for a reasonable window for the selection of coating materials and preparation techniques.

In the concept of coated ATF claddings, the coating material is an important factor that determines the performance of the cladding. As mentioned previously, the most conventional and effective high-temperature protective coatings are based on the formation of $Cr_2O_3$, $Al_2O_3$, and $SiO_2$ continuous protective layers with certain scales. This means that any ATF coating should contain at least one Cr, Al, or Si element. Table 3 presents some coating materials for which applications for ATF claddings has recently been proposed.

**Table 3.** Candidate materials for ATF cladding coatings [13,14,31,46,70,71,87].

| Type | Potential High-Temperature-Resistant Coating Materials |
| --- | --- |
| Metals | Cr, CrAl, AlTiCr, FeCrAl, 310SS, and high-entropy alloys |
| Carbides | MAX phase carbides ($Cr_2AlC$, $Zr_2AlC$, $Ti_2AlC$, and $Ti_3SiC_2$), $Cr_xC_y$, and SiC |
| Nitrides | CrN, AlCrN, TiAlN, TiAlCrN, and TiAlSiN |
| Silicides | $ZrSi_2$ |
| Multilayer | Cr-Zr/Cr/CrN, CrN/Cr, Cr/CrAl, Cr/FeCrAl, Mo/FeCrAl, and TiN/TiAlN |

Generally, $Cr_2O_3$, $Al_2O_3$, and $SiO_2$ layers cannot protect against temperatures higher than their maximum temperature thresholds of 1000–1100, 1400, and 1700 °C during the long-term ($\gg$1000 h) service life under isothermal conditions, respectively [88]. To maximize the safety margin of a reactor by improving the resistance of ATF coating claddings to high-temperature steam oxidation, it is expected that one use $Al_2O_3$ and $SiO_2$ coating materials. However, previous studies have shown that when these two oxides are used individually, they exhibit instability in the chemical environment of a water-cooled reactor [89,90], namely hydration and dissolution occur and are accelerated, especially under an irradiation environment [90]. To address this issue, the use of water-resistant corrosion coatings has been suggested to completely reduce the dissolution of Al-containing and Si-containing coatings. Alat et al. [91] developed a TiN/TiAlN multilayer coating with a 1 µm-thick TiN top layer on the coating, and they confirmed that the dense $TiO_2$ formed from TiN in a hydrothermal environment can act as an effective barrier to Al out-diffusion. However, Ti rapidly oxidizes at high temperatures and diffuse in the coating, which may weaken the protective effect of $Al_2O_3$ or $SiO_2$. Pint et al. [51] found that $Ti_3SiC_2$ failed to form a protective $SiO_2$ layer during high-temperature oxidation. In addition, Tunes et al. [92] observed that TiN films are prone to dissociate under energetic $Xe^+$ ion irradiation and to form a Ti-rich region which, similar to Zr, leads to a large hydrogen gas release during high temperature oxidation. Therefore, they believe that TiN is not suitable for ATF. Compared to the above-mentioned Al-containing coatings, the dissolution of FeCrAl alloy and CrAl alloy coatings that contain Al has not been observed. According to Kim et al. [34], the improved corrosion resistance of FeCrAl alloy (Fe-22Cr-5Al-3Mo) and CrAl alloy (Cr-15Al) coatings under normal operating conditions can be attributed to the formation of a stable Cr-rich spinel structure, which suppresses Al diffusion.

Though FeCrAl coatings are stable under standard operating conditions of water-cooled reactors, the diffusion of metals and the subsequent formation of eutectics at high temperatures (>900 °C) still represent serious issues. Terrani et al. [93] prepared two outer protective layers made of an FeCrAl alloy and 310SS steel on the surface of a Zr alloy via hot isostatic pressing (HIP). Their degradation at 1300 °C was caused by metal diffusion, which induced the formation of a layer with a thickness of several hundred micrometers at the interface between the coating and the matrix. Zhong et al. [94] deposited FeCrAl coatings with different compositions on a Zr-2 alloy using magnetron sputtering, which substantially reduced the oxidation rate of the cladding in the steam environment at 700 °C. However, an Fe-Zr eutectic reaction was observed at ~900 °C. Oxidation temperatures higher than 900 °C are known to induce rapid film degradation. The similar behavior of an FeCrAl coating on a Zr alloy prepared via the cold spraying method was found by Park et al. [95] after exposure to high-temperature steam oxidation at 1200 °C for 3000 s. It can be seen that the diffusion of metals and eutectic reactions substantially limit the inherently excellent high-temperature oxidation resistance of FeCrAl. An Mo layer of a certain thickness deposited between FeCrAl coating and Zr matrix acts as a diffusion barrier that mitigates or prevents the above-mentioned problems [73,95]. However, further experimental research is still needed to verify the performance and reliability of Zr alloys coated with Mo and FeCrAl.

CrN coatings have been found to exhibit good performance under normal operating conditions [90,96], as well as under conditions specified in high-temperature steam oxidation tests [19]. Additionally, R.V. Nieuwenhove et al. [90] confirmed the outstanding stability of a CrN coating (with a thickness <5 µm) on the surface of a Zr alloy under typical fuel irradiation conditions using in-reactor

testing. However, according to the findings from Terrani et al. [46], recent results of LOCA tests on CrN-coated Zr alloy cladding showed that although the CrN coating was well-attached to the matrix without any apparent delamination, the ceramic coating could not maintain the circumferential extension on the ballooning area of the cladding. This yielded the formation of many surface cracks due to inherent brittleness of the ceramic, limiting the ability of the coating to protect the cladding from high-temperature steam oxidation [14], which resulted in a comparable oxidation or bursting behavior of the coated and the uncoated claddings (Figure 4 [46]).

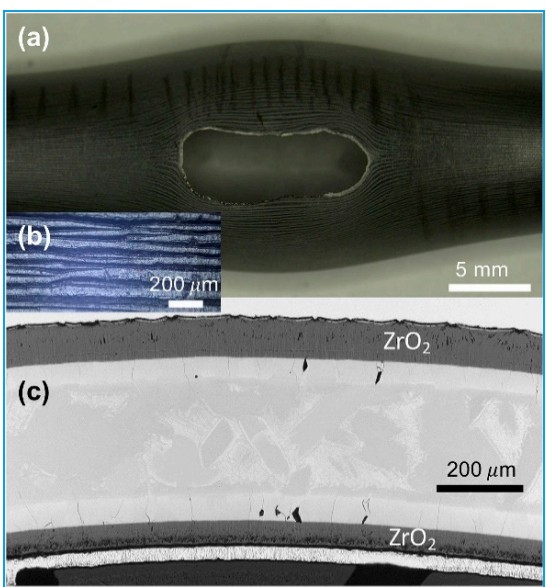

**Figure 4.** Results of a LOCA test on a CrN-coated Zr alloy cladding: (**a**) balloon and burst area, (**b**) cracks on the coating surface, and (**c**) cross section of oxidized CrN-coated Zr alloy. Reproduced from [46] with copyright permission from Elsevier.

MAX phase carbides ($Cr_2AlC$, $Ti_2AlC$, and $Ti_3SiC_2$) have also been proposed for ATF coatings. However, recent studies have shown that their performances are not so satisfactory. Roberts [97] reported that Ti-Al-C coatings produced multiple oxide and hydroxide phases, and much of the Cr-Al-C coatings spalled off the substrate during corrosion tests. Ang et al. [85,86] conducted a neutron irradiation test on a MAX phase material in an LWR environment. The test results indicated the anisotropic swelling, apparent cracking, and poor mechanical strength of the Ti-Al-C series; although the Ti-Si-C materials showed a relatively better radiation resistance than Ti-Al-C, but a high reduction in the content of $Ti_3SiC_2$ to TiC occurred after irradiation. Additionally, Tunes et al. [98] found that both the two Ti-based MAX phases underwent phase decomposition and irradiation-induced segregation under neutron irradiation at 10 dpa. The amorphization of $Cr_2AlC$ thin films was observed at low dose under room temperature ion irradiation [99,100]. Though the $Cr_2AlC$ thin film showed good irradiation resistance at 623K [100], its high irradiation hardness [100] may increase the risk that coating cracks and losses its protective ability when the cladding has large strain, as occurred with CrN coatings during the simulated LOCA transient tests [46]. In addition, the preparation of MAX phase coatings is also a problem. It is difficult to obtain a pure MAX phase at a low temperature, and the microstructure of a zirconium alloy matrix is prone to change at high temperatures [97,101–103]. Overall, MAX phase materials are not so suitable as coating materials for ATF claddings in an LWR environment.

Metallic chromium (Cr) has a high melting point, good resistance to high-temperature oxidation, intrinsic ductility at high temperatures, and a coefficient of thermal expansion that is similar to Zr [13,16]. These characteristics qualify Cr as a coating material for Zr alloy claddings. The mechanical strength of pure Cr does not extensively differ from that of industrial Zr alloys (from room temperature to the cladding's service temperature of approximately 350 °C), and a thin outer Cr coating has a limited

effect on the overall mechanical properties of the cladding under normal operating conditions [14,17,69]. Due to the good hardness and strong adhesion of the Cr coating, Cr-coated Zr alloy claddings have shown substantially improved wear resistance compared to uncoated claddings [14,18]. This would greatly reduce the probability of cladding failure due to grid-to-rod vibration and debris friction during reactor operation. In an LWR cooling water environment, Cr coatings have been found to significantly improve the corrosion resistance of Zr alloy claddings [14,18–20,104], which is beneficial to the life extension and burnup promotion of a nuclear fuel element. Additionally, a Cr coating was found to enhance the chemical resistance of the Zr alloy to an environment of primary cooling water, which improved the flexibility of operations under current water chemistry limitations [18]. The results of ion irradiation and neutron irradiation tests have shown that Cr coatings with body centered cubic (BCC) structures have a certain irradiation stability [105,106]; those prepared via the cold spraying method, especially, exhibited irradiation damage resistance [107]. The overall assessment of Cr coatings in ex-situ conditions, i.e., out of a reactor, confirmed their good adhesion, which helps to maintain the integrity of the coating during normal operations.

In terms of accident resistance, Cr coatings have been found to demonstrate outstanding resistance to high-temperature steam oxidation. The oxidation rate of a coated Zr alloy cladding is lower by at least one order of magnitude compared to an uncoated cladding [21,22,73,104,108,109]. Furthermore, Cr coatings greatly reduce the hydrogen absorption of claddings, alleviating the risk of hydrogen-induced embrittlement. This, in turn, enables a cladding to maintain its strength for a longer time and doubles the oxidation time before the occurrence of cladding failure during quenching [104,108,109]. The introduction of a Cr coating is expected to raise the peak cladding temperature (PCT) from 1200 °C, as specified in the current standard, to 1300 °C [46], while reducing the release of flammable hydrogen. Additionally, a Cr coating imparts a certain degree of strengthening to the general high-temperature mechanical response of Zr alloy cladding [17,95,110–112]. This strengthening effect can be explained from two aspects: Cr has higher intrinsic resistance to high-temperature creep than Zr [110], and a Cr coating improves the oxidation resistance of a Zr alloy in high-temperature steam and thus reduces the hydrogen-pickup so that the Zr matrix retains a higher residual strength, toughness, and ductility. This strengthening effect reduces the high-temperature creep rate of a Zr alloy cladding and increases the bursting time in the early LOCAs by two-to-three times [110,111]. Meanwhile, the bursting temperature is increased to a certain extent, and the sizes of the balloon and the rupture opening also become smaller [111,112]. These changes can mitigate the situation of a blocked coolant channel in a nuclear fuel sub-assembly. Especially when a small-sized rupture (only 1 mm$^2$) is generated after the bursting of the cladding that occurs at a temperature higher than 850 °C (the temperature range for the coexistence of $\alpha_{Zr}$ and $\beta_{Zr}$ or a complete $\beta_{Zr}$ phase) [110–112], the migration and diffusion of fuel under high burnup is mitigated. Moreover, by reducing steam intrusion into a cladding (in the gap between the cladding and the core), the oxidation degree of the inner surface of the cladding and the accompanying secondary hydrogenation are limited [14]. Additionally, even in the vicinity of severely deformed bursting areas, a Cr coating still maintains its adhesion to its matrix and a high degree of integrity, thus continuing to generate a protective effect during LOCAs. Overall, a Cr coating can significantly improve the stability and integrity of Zr alloy claddings during a LOCA, providing more "processing time" for an operator and improving the passive safety of the reactor.

It is worth mentioning that besides the constant attention paid to the research and development of Cr coatings, the Korea Atomic Energy Research Institute (KAERI) of South Korea has creatively proposed the concept of an ATF cladding with a modified surface that integrates Cr or CrAl coatings and an oxide dispersion-strengthened (ODS) surface treatment [6,33,34,69]. The aim was to simultaneously improve high-temperature steam oxidation resistance and high-temperature mechanical properties of Zr alloy claddings. The results in their out-of-reactor performance research showed that a CrAl coating exhibited dual excellent performance under normal and accident conditions; the creep resistance of a Zr alloy cladding subjected to the surface ODS strengthening treatment at 380 °C and its anti-ballooning/bursting performance in a LOCA simulation test were both notably improved

compared to the results obtained for the coating without ODS treatment [34]. Additionally, results from the tests in the Halden Research Reactor also showed that these ATF claddings with modified surfaces could operate stably without any signs of failure [33]. This further verified the outstanding performance of Cr-based coatings.

Table 4 [46] summarizes the key performance characteristics of the most-proposed candidate materials for ATF coatings based on the above discussion. It can be seen that Cr-based coatings have been found to exhibit excellent performance in all relevant aspects. Thus, they have become the most promising materials for ATF fuel claddings.

**Table 4.** Key performance characteristics of the most proposed candidate materials for ATF coatings. Adapted from [46] with copyright permission from Elsevier.

| Key Performance | Cr | TiAlN | TiN/TiAlN | FeCrAl | CrN | $Ti_2AlC$ | $Ti_3SiC_2$ | $Cr_2AlC$ |
|---|---|---|---|---|---|---|---|---|
| Corrosion behavior | √ [14] | × [91] | √ [91] | √ [34] | √ [96] | × [97] | - | × [97] |
| Irradiation behavior | √ [104] | - | × [92] | - | √ [90] | × [98] | × [85] | × [100] |
| Oxidation behavior | √ [14] | √ [113] | - | × [93] | × [46] | √ [73] | × [51] | √ [114] |

## 5. Preparation of Cr-Based Coatings

At present, the most extensively used preparation methods of the Cr-based coatings on Zr alloys are physical vapor deposition (PVD) [14,18–20,22,23,55,104–106,108,109,115,116], cold spraying (CS) [15,24,73,95,107,117–119], and 3D laser melting and coating (3D-LMC) [6,17,25,33,34,69,120]; each are currently used by CEA (French Alternative Energies and Atomic Energy Commission) in France, Westinghouse Electric Corporation in the US, and KAERI in South Korea, respectively. These preparation methods and the characteristics of the prepared coatings are described in the next section.

### 5.1. Physical Vapor Deposition

The most widely used and fastest-growing coating technology is based on the vapor deposition method, which can be further divided into chemical vapor deposition (CVD) and PVD. Compared with CVD, PVD can lower the deposition temperature of a matrix to below 500 °C. This coating technology has been used in the industrial manufacturing of hard coatings and functional thin-film materials for aerospace, electronic communication, energy, chemical engineering, and many other applications [121].

In PVD technology, a certain physical method is used to excite gaseous particles (atoms, molecules, or partially ionized particles) on the surface of raw materials under vacuum. Then, the deposition of thin films on the substrate is achieved through the free movement of these particles or their acceleration in the electric field. Based on the excitation method used to produce thin-film materials, PVD can be divided into three sub-categories: evaporation plating, sputtering plating, and ion plating. In the evaporation plating method, particles that form a thin film are generated by heating of raw materials in the molten pool until evaporation at high temperatures; in the sputtering plating method, raw materials (targets) are used as electrodes, and sputtered particles are obtained by bombarding the target surface with high-energy electric field-accelerated plasma particles generated by glow discharge of Ar gas between the electrodes; in the ion plating process, which was developed based on the evaporation and sputtering methods, target atoms are evaporated and ionized by low-pressure discharge—such ions with extremely high incident energy are obtained via the application of a negative bias to the matrix. The PVD method has the following advantages: the realization of low-temperature (down to room temperature) deposition, flexible and controllable parameters in the processing procedures that facilitate the control of the coating quality, and a high probability to produce dense coatings, smooth surfaces, favorable mechanical properties, and a good adhesion of the coating on the matrix.

However, the PVD method has some disadvantages: The coating of large samples is complicated, it has a low deposition rate (usually from 1 to 5 μm/h), and it has a line-of-sight deposition that hinders the deposition of uniform coatings on complex three-dimensional workpieces [121–123].

The desired thickness and microstructure of coating can be obtained by the optimization of process parameters, such as deposition temperature, current or power, gas pressure, and bias. Figure 5 demonstrates a typical PVD process and cross-sectional morphology of a prepared Cr coating. A dense tightly-bound film with the columnar grain structure perpendicular to the surface plane of the matrix can be observed without apparent microstructural defects [14]. The French CEA Institute continually develops Cr coatings by the PVD method. Ribis et al. [124] performed an atomic scale characterization and analysis of the interface between a Cr coating prepared by the PVD method and a Zr matrix. The results showed that the Cr/Zr interface was composed of nanoscale Zr $(Cr,Fe)_2$ intermetallic compound sublayers, and continuity between different crystal phases could be observed within an area of several microns. These phenomena confirm the characteristics of local heteroepitaxy, ensuring adequate bonding strength of the coating. Additionally, Wu et al. [105] examined the Cr/Zr interface of a PVD Cr-coated (2.5 μm) Zr alloy by irridiation at 400 °C with 20 MeV $Kr^{8+}$ ions, and the calculated displacement damage at the Cr/Zr interface was 10 dpa. The interface retained the continuity of the lattice structure after irradiation, indicating a good microstructural stability of the Cr/Zr interface. Ribis et al. [125] conducted a tensile test of a Cr-coated Zr-4 alloy exposed to neutron irradiation at 350 °C. The corresponding fracture analysis showed that after 2 dpa of irradiation, the Cr coating did not delaminate and still possessed residual adhesion performance.

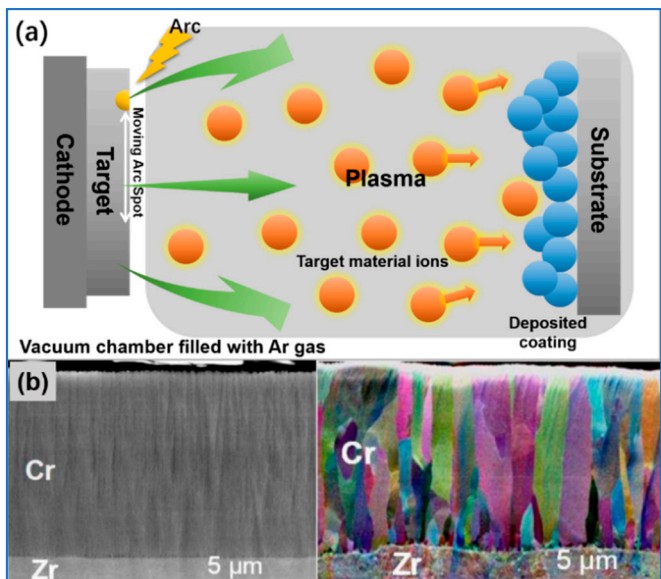

**Figure 5.** (**a**) A schematic diagram of a typical physical vapor deposition (PVD) process and (**b**) the cross section of the prepared coating. Reproduced from [14] (**b**) with copyright permission from Elsevier.

Brachet et al. [14] studied the room-temperature tensile properties of the Cr-coated Zr-4 specimens obtained by the PVD method. Figure 6 shows that a Cr coating with a thickness of 10–15 μm did not significantly influence the tensile properties of a Zr-4 sample. Some cracks appeared at the fracture of the Cr coating due to the intrinsic low-temperature brittleness of Cr, but even after the fracture, Cr fragments still adhered to the matrix, thus confirming the good bonding strength of the Cr/Zr-4 interface. However, the mechanical properties of the Cr-coated Zr-4 samples prepared under unconventional process conditions (at temperatures higher than the recrystallization temperature of the Zr alloy) were notably changed. Therefore, deposition methods achieved at low temperature that

can provide low damage deserve extensive attention, and among them, the PVD and CS approaches have considerable advantages for the deposition of coatings on the outer surface of Zr alloy claddings.

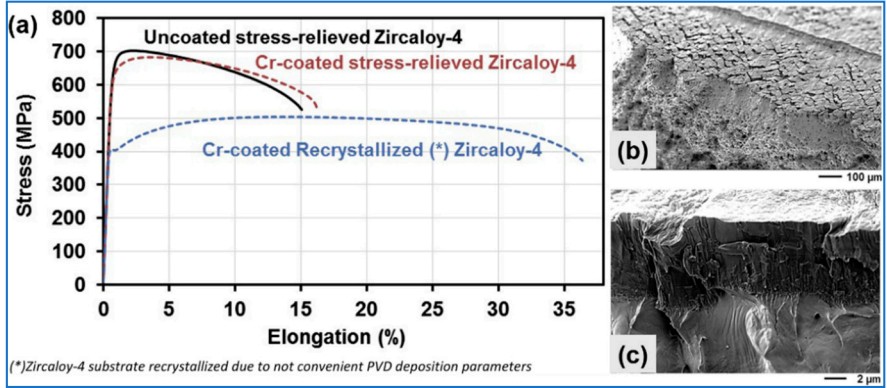

**Figure 6.** Results of tensile tests at room temperature (RT) on the Cr-coated Zr-4 sheets prepared by the PVD method: (**a**) engineering tensile curves; (**b**,**c**) SEM fractographs. Reproduced from [14] with copyright permission from Elsevier.

### 5.2. Cold Spraying

CS technology is a low-temperature powder spraying process developed from traditional thermal spraying more than a decade ago. Compared with traditional thermal spraying technology, powder particles with sufficient energy in the CS method are generated by high acceleration rather than by high-temperature melting. At low temperatures, powder particles do not undergo phase transformations, such as oxidation, decomposition, and evaporation, during the deposition process, thus avoiding the extensive formation of oxide inclusions, pores, and phase separation in coatings [126].

In the preparation process, the powder particles of the coating material were firstly mixed with a preheated inert gas under pressure (nitrogen or helium) in a spray gun and then accelerated and ejected at supersonic speed (200–1200 m/s) to hit the surface of a matrix. The solid deposition of the coating is realized by the high-strain-rate plastic deformation and a related adiabatic shear process among the particles, as well as between the particles and the matrix. This yields mechanical interlocking and metallurgical bonding between the particles and the matrix surface [119]. The desired thickness and microstructure of a coating can be achieved through the optimization of process parameters such as the gas composition, gas preheating temperature and pressure, spraying distance, particle size, cladding rotation speed, and translational velocity of the spray gun. A typical schematic diagram of the CS process and corresponding cross-sectional morphology of a CS coating are presented in Figure 7 [15]. It can be seen that a CS coating differs from a PVD coating in terms of thickness uniformity, interface roughness, and other morphological characteristics.

The CS technology has two main advantages: a high deposition rate (>10 μm/min) and the possibility to operate at room temperature and atmospheric pressure. Therefore, from the economic and industrial scaling-up perspectives, CS technology is more likely to be extended to commercial-scale manufacturing of full-size coatings on the surface of fuel claddings. However, CS technology has also some shortcomings, which can be summarized as follows: an excessively high deposition rate is challenging for the quality control of coatings; the plastic deformation and work hardening of the matrix surface are more likely to occur due to the inherent particle collision mechanism in the spraying process; the quality of the coating depends to a great extent on the characteristics of raw powders, thus increasing the requirements in that direction; and the excessive thickness of the coating and high roughness of the surface may invoke further polishing to optimize these parameters, i.e., to reduce the coating thickness to an appropriate level and to obtain a surface roughness closer to that of Zr alloy claddings [73,117].

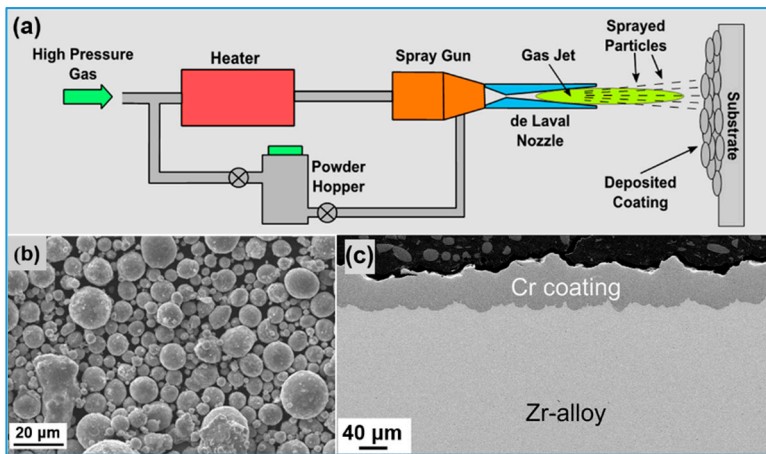

**Figure 7.** (**a**) A schematic diagram of a typical cold spraying (CS) process; (**b**) sprayed particles; and (**c**) a cross section of the sprayed coating. Reproduced from [15] with copyright permission from Elsevier.

Shah et al. [72] from Westinghouse Electric Corporation investigated the stress–strain characteristics of three cold-sprayed Cr-coated (~50 μm) OPT-ZIRLO Zr alloy cladding tubes (unpolished) by streching them to 1%, 10% strain and fracture failure, respectively, at room temperature. The results did not show any signs of delamination, even in the necked area of the failed sample under very large strains. The yield strength and ultimate tensile strength of the failed coated sample were similar to the uncoated sample (the differences between the two indexes were both within 3%). Though the total elongation of the coated samples was lower than that of the uncoated control tubes, the elongation results fully met the specifications of the OPT-ZIRLO cladding material. These results indicated that a CS Cr-coating with a thickness of 50 μm can maintain its structural integrity at a large strain, and the coating itself has a minor impact on the mechanical properties of a Zr alloy matrix.

Maier et al. [107] characterized a CS Cr coating using a TEM in-situ ion irradiation test. The as-deposited sample (Figure 8a) contained elongated grains that underwent severe plastic deformation during the CS process, and each grain exhibited a dense dislocation network structure formed through local dynamic recrystallization; the annealed samples (cut out from the same sample and annealed at 800 °C for 8 h; see Figure 8) contained more equiaxed grains and fewer deformation defects because the deformation effect during the CS process was eliminated. After the exposure of two samples to 1 MeV of $Kr^{2+}$ irradiation at an ion flux of $9.6 \times 10^{-4}$ dpa/s at 320 °C, the irradiation damage gradually increased to 3.0 dpa; irradiation-induced defects in the annealed samples were generated at an earlier stage than the as-deposited sample, and the size and density of these defects were larger (Figure 8c,d). The above results confirmed that in a Cr coating material that undergoes severe plastic deformation, the pre-existing deformation-induced defects act as traps to resist irradiation-induced damage, thus yielding a better irradiation stability.

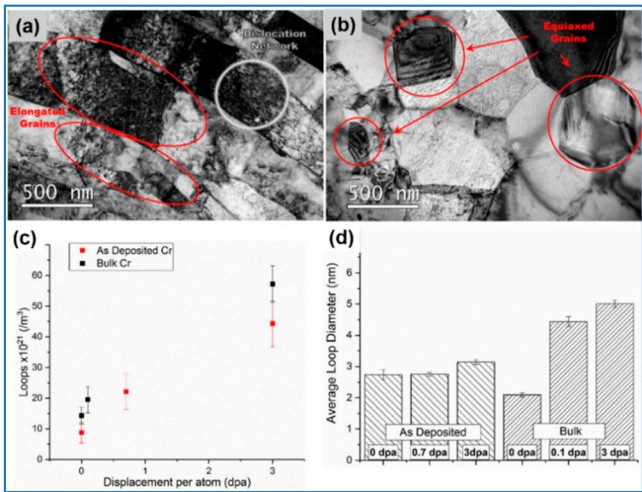

**Figure 8.** Grain morphology of (**a**) as-deposited and (**b**) annealed Cr coatings prepared via the CS method. The (**c**) density and (**d**) size of irradiation-induced defects as a function of radiation damage. Reproduced from [107] with copyright permission from Elsevier.

### 5.3. 3D Laser-Melt Coating

As an emerging deposition technology, 3D-LMC has been applied for the preparation of Cr coatings on the surface of Zr alloy claddings by KAERI of South Korea [6,17,25,33,34,69,120]. In a typical 3D-LMC preparation process, the laser beam generated by a laser source passes through a parallel movable laser head and then focuses on the surface of a Zr alloy substrate. The particles of Cr powder, along with an inert protective carrier gas (argon), are supplied to the center of the laser beam-irradiated spot on the surface of the Zr alloy cladding through a power supply system. After the melting and solidification of the particles, a continuous metallurgical coating is formed on the surface scanned by the laser beam.

Compared with CS coatings, 3D-LMC coatings have denser microstructures and higher bonding forces. However, a wide (100–200 μm) heat-affected zone (HAZ) is formed on the surface of the Zr alloy matrix next to the coating [17]. Figure 9 demonstrates the 3D-LMC process and the characteristics of a coating prepared with it. It can be seen that the Cr coating prepared by this method was tightly coupled with its Zr alloy substrate, without pores, cracks, or traces of oxidation in the coating. A gradient of the composition formed at the coating/substrate interface. However, the coating had a rough surface due to the adhesion of molten Cr particles during the cladding process. The change of microhardness was not directly related to the thickness of the Cr coating, but it was related to the formation of the HAZ in the Zr alloy matrix. Therefore, coatings with more evenly distributed thicknesses and smaller HAZ area ranges can be obtained through the optimization of process parameters, such as laser power, scanning speed, airflow rate, and powder flow rate.

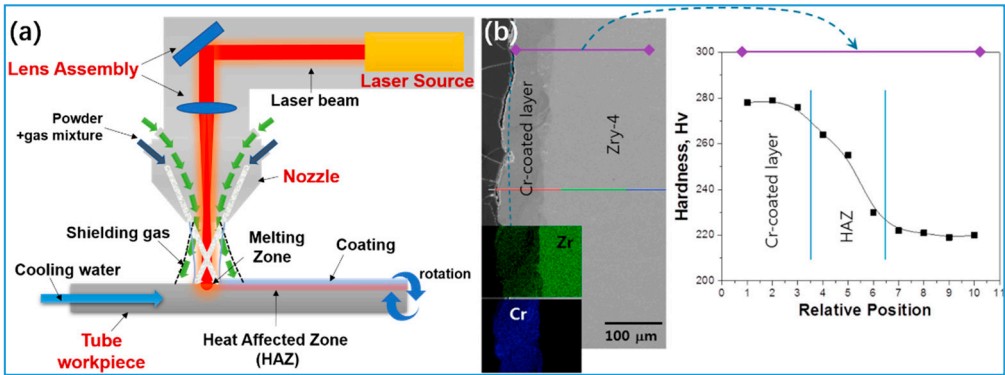

**Figure 9.** (**a**) A schematic diagram of a typical 3D laser melting and coating (3D-LMC) process and (**b**) characteristics of the prepared coatings. Reproduced from [17] (**b**) with copyright permission from Elsevier.

3D-LMC and CS coating technologies exhibit a high deposition rate and can be operated under an atmospheric environment; however, the former is not as flexible as the latter. Additionally, both methods have similar disadvantages, such as the requirement for further polishing treatment due to the formation of excessively thick coatings and rough coating surfaces, as well as a low level of coating quality control due to their high deposition rates. Additionally, the influence of powder characteristics on coating quality is considerable in terms of the thickness, density, microstructure, and composition of the interdiffusion area. For instance, if the Cr powder is not properly supplied to the center of the laser beam-irradiated area on the matrix surface, the obtained Cr coating on the matrix surface may exhibit a low uniformity level. Therefore, 3D laser melting technology raises more rigorous requirements for the control of powder characteristics. Apart from the formation of an HAZ, the deformation of the Zr alloy cladding tube may also occur during the laser heating process because extremely high thermal energy is instantly generated when the laser beam hits the cladding surface. Therefore, cooling the cladding tube is a very important step in the coating process. In the actual operating process of 3D-LMC, cooling water continuously flows through the inner surface of the cladding tube to minimize the negative impact of the temperature rise on the cladding tube [104].

To verify the ability of the coating to maintain integrity during the hoop deformation of the coated cladding tube, Kim et al. [17] cut off a 2 mm-wide ring sample from a Cr-coated Zr-4 cladding tube (with a coating thickness of 80 μm) prepared by the 3D-LMC method, and they performed ring tensile and compression tests (at a strain rate of 1 mm/min). It can be seen from Figure 10 that the hoop strength of the Cr-coated sample was better than that of the uncoated sample, which may be explained by the increased coating thickness (approximately 80 μm) or by the formation of a rapidly solidified structure (such as martensite) in the 3D LMC process, which improved the hoop strength of the cladding. The delamination of the coating was not observed, but the Cr coating completely cracked in severely deformed areas. A series of ring tensile tests with strains of 2%, 4%, and 6% on Cr-coated cladding samples was conducted to confirm the ultimate strain at the coating failure. The results showed that the Cr-coated cladding could withstand a 4% strain without cracking. When the strain reached 6%, a series of radial cracks formed in the Cr coating. The ultimate strain of the coating still met the requirements of a 1% hoop strain resistance according to the fuel cladding specifications.

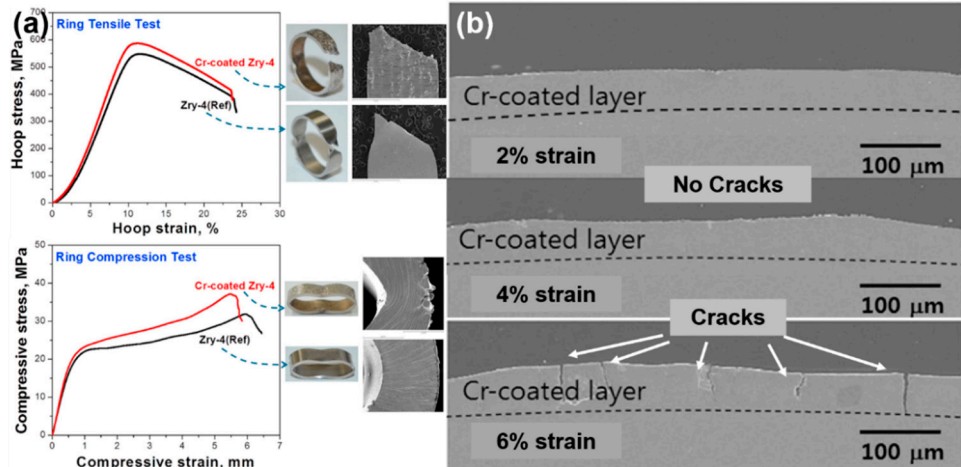

**Figure 10.** Results of ring tensile and ring compression tests at RT on Cr-coated Zr-4 cladding samples prepared by 3D LMC: (**a**) engineering curves with corresponding photos of failure samples and (**b**) cross-sectional SEM images of Cr coating at different strains. Reproduced from [17] with copyright permission from Elsevier.

Additionally, researchers from the China Nuclear Power Technology Research Institute (CNPRI) [21] prepared a Cr coating on the outer surface of a Zr alloy cladding using air plasma spraying (APS), which is a thermal spraying method. Though the obtained coating contained oxide inclusions, it improved the high-temperature steam oxidation resistance of the Zr alloy cladding to a large extent. Wei et al. from Nuclear Power Institute of China (NPIC) [20] chose the multi-arc ion plating (MAIP) method, which is more cost-effective and has a better coating adhesion than the PVD method, to prepare Cr-coated Zr alloys, and the resistance of the coated samples to hydrothermal corrosion and high-temperature oxidation was significantly improved compared to the uncoated control samples.

The risk that the inner surface of nuclear fuel cladding will undergo high-temperature oxidation is also a realistic scenario because the steam in the outer environment enters the cladding tube through a rupture that opens after cladding balloons and bursts under LOCA conditions, thus causing the oxidation and secondary hydrogenation of the cladding's inner surface and complicating the accident situation further. To overcome the challenges of the formation of a uniform coating on the inner wall of a long and narrow tube, CEA recently collaborated with some French universities to develop a special metal–organic chemical vapor deposition (MOCVD) process [127–131]. This method can provide a coating of a Zr alloy matrix at temperatures lower than the recrystallization temperatures of the Zr alloy. At present, they have successfully plated a Cr-based coating on the inner surface of a cladding tube. However, many grain boundary gaps, as well as impurities or metastable phases, formed in the polycrystalline Cr coating prepared by this method. The high-temperature oxidation resistance of the Cr-based coating failed to match that of the $Cr_xC_y$ amorphous coating prepared by the same MOCVD method because the amorphous material did not possess grain boundaries in the microstructure and with that, no channels for the rapid in-diffusion of oxidants through the coating [131].

Though the preparation of a Cr alloy coating can be achieved by the above-mentioned methods, a pure metallic raw material should be replaced with a raw alloy material. Thus, an alloy target is required in the PVD method, while an alloy powder should be used to form the coating in the CS and 3D LMC methods. Additionally, the preparation of a Cr alloy coating can also be achieved by controlling the power or current of multiple pure metallic targets during the PVD process. Currently, some researchers are performing research on this topic [132–136]. However, it should be noted that alloy coatings prepared by the PVD method tend to undergo the segregation of the coating/target material components [137], resulting in a compositional differences between the coating and the

alloy target. This is undoubtedly a huge challenge for the accurate control of the composition of an alloy coating.

## 6. Summary and Prospect

In summary, to improve the performance of Zr alloys, the ATF concept has been proposed. Zr alloys with Cr-based coatings have been found to exhibit excellent service performance under both normal and accident operating conditions, which led to the nomination of this topic as a research hotspot due to its promising engineering applications. After the literature review on this topic, it can be concluded that research institutes from different countries have proposed their representative processes for the preparation of Cr-based coating on Zr alloys. These mainstream processes include PVD, CS, and 3D-LMC. Though the characteristics of coatings prepared by different processes differ from one to another and the thickness of the coating varies within a wide range from 5 to 80 μm, in general, coatings prepared using these three preparation methods are well-integrated with Zr matrixes. Additionally, a coated cladding can withstand a certain mechanical load without the apparent irradiation-induced degradation of its coating–matrix interface. All these features are the key to reliable coatings.

Though it has been confirmed that some properties of Zr alloys with Cr-based coatings are better than those of standard uncoated Zr alloys, Cr coatings prepared by different methods or even different processes of the same method may result in very different performance levels. This indicates that there is still space for the optimization of the Cr-based coating technology, including the improvement of the coating's intrinsic characteristics, such as structure, defects, and thickness. The literature that will be presented in Part II of this review will specify the effects of these coating characteristics on the performance and degradation behavior of coatings, but the mutual relationship has yet to be systematically clarified. Additionally, since the coating will adversely affect the neutron economy of a reactor, the corresponding countermeasure for this issue is to minimize the coating thickness as much as possible. The performance loss caused by the minimization of the coating's thickness can be compensated for by the addition of alloying elements or by a more beneficial structural design. Unfortunately, to date, there have been no satisfying results. Therefore, to achieve an optimal coating design and preparation process, extensive further research on this topic has to be performed, and some pathways toward this goal can be summarized as follows: First, the correlation between the intrinsic characteristics of a coating and its application performance must be established; second, the key factors that lead to the accelerated degradation or failure of the coating must be determined; and third, the balance between performance and economic efficiency must be considered.

**Author Contributions:** Investigation, H.C.; writing—original draft preparation, H.C.; writing—review and editing, R.Z. and X.W.; supervision, X.W. and R.Z. All authors have read and agreed to the published version of the manuscript.

**Funding:** This research was funded by State Administration of Science, Technology and Industry for National Defense, PRC, grant number "20181722".

**Acknowledgments:** The review presented was supported by the Research on Key Technology of Accident Tolerant Fuel of the Nuclear Energy Development Project in the State Administration of Science, Technology and Industry for National Defense, PRC.

**Conflicts of Interest:** The authors declare no conflict of interest.

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
