# Peer review of "Application and Development Progress of Cr-Based Surface Coatings in Nuclear Fuel Element: I. Selection, Preparation, and Characteristics of Coating Materials"

_coatings, doi:10.3390/coatings10090808_

Round 1
Reviewer 1 Report
Dear Authors,
I have stopped to review your paper in section 5, page 10. The reason for this is that a major flaw is the lack of appropriate referencing in several paragraphs. A lot is said, but no references are added to support statements which is not acceptable for a paper that has the intention of reviewing the latest research in the field of ATFs.
For clarification and due to the short time given to review, most of my comments were written directly onto the paper; the PDF file is attached to my revision.
I would like to give you a second opportunity to revise the paper, so I can review it properly.
For example:
- The introduction has zero references;
- Table 1: a lot of informations, but no referencing supporting the statements. I understand the table has been reproduced from reference [10], but you have to provide your input on it.
- On table 2: It is of knowledge from this reviewer that ONRL also studied and considered TiN coatings for ATF fuels. The irradiation response of TiN coatings was also investigated.
- On table 3: it has been mentioned some candidate coatings which are potential for ATF applications. However, not even a single reference is provided.
- On table 4: TiN has been investigated under irradiation, but the table has not been updated. See my comment in point 3 and the reference on it.
- On table 5: what about the irradiation resistance of Ti-based MAX phases under neutron irradiation? Are MAX phases suitable for nuclear reactors? This reviewer is aware of the limited number of publications on the neutron irradiation behaviour of MAX phases, but this is not covered in the paper.
- On table 5: Cr2AlC is a potential coating for ATF and not considered by authors. The radiation resistance of such coatings is a subject of intense research most recently.
- A lot is said about H into Zircaloys, but no classical reference or paper is discussed in the text. Are your future ATF coatings be subjected to H embrittlement and formation of hydrides?
I would be delighted to read a second time.

Reviewer 2 Report
A very well written article containing a valuable analysis of coatings used in the nuclear industry. I recommend it for printing without corrections.
Author Response
Thank you very much for your kindness and recognition!
Reviewer 3 Report
This paper is new, original and well organized. English language is good in the paper and all references are adequate. Also all parts of paper are important and conclusion is fine. I recommend this paper for publication.
Author Response

(The authors gave the same response as above.)

Reviewer 4 Report
Dear Editor
I believe that this manuscript is a valuable contribution to the literature. I thus recommend publication, but only after my comment is addressed.
With my best regards
Report to Coatings.
The article review focused on accident tolerant fuel (ATF), which is an interesting topic mostly after the Fukushima nuclear accident in Japan. This attack triggered the shifting of attention to the safety of nuclear power plants around the world and promoted the development of new nuclear technologies. The ATF was designed to improve the passive safety property of the water-cooled reactor, increasing the safety allowance, and reducing the dependence of the reactor on active cooling. In the beginning, the authors give a summary of nuclear power plants and they mention the importance of the Cladding which is one of the essential
components that facilitate the fuel elements to remain intact and ensures the safety of the nuclear power plant. The used cladding material is Zirconium alloy cladding. Then the authors focused on the proposed developing method which makes the cladding more safe. The authors introduce the efficiency of coated ATF claddings and they mention some coatings as Cr, CrAl, AlTiCr, FeCrAl, 310SS, and high-entropy alloys…. Among these coatings, The CrN coating exhibited good performance under normal operating conditions, as well as
276 under conditions specified in high-temperature steam oxidation tests. Then the authors detailed the different methods of the preparation of Cr-coating.
Overall, I believe that this manuscript is a valuable contribution to the literature. I thus recommend publication, but only after my comment is addressed.
The authors mentioned in lines 184 to 187 that two major technical routes have been established around the world. One route is to develop alternative materials, such as SiCf/SiC composites, FeCrAl stainless steel, and Mo alloy, which can completely replace zirconium alloys. The other route is to prepare coatings on the surface of the existing zirconium alloy claddings.
Could the authors explain in a few sentences which one is safer and the reasons could be based on the irradiation damage of neutrons and light gas (He and H) that create bubbles and/or blister causing embrittlement, creep, fracture… based on these references:
FeCrAl: (Journal of Nuclear Materials 465 (2015) 746-755; Journal of Nuclear Materials 483 (2017) 21-3422 ).
SiC: (Acta Materialia 188 (2020) 609-622 ; Acta Materialia 181 (2019) 160- 172; Journal of the European Ceramic Society 38 (2018) 1087–1094 ).
Mo alloys: (Journal of Nuclear Materials 376 (2008) 240–246; Nuclear Engineering and Technology 48 (2016) 16-25).
Round 2
Reviewer 1 Report
The paper has been improved and it is a good contribution to community.